# How do gender norms contribute to stunting in Ntchisi District, Malawi? A qualitative study

**Whitney Mphangwe[1,2], Ann Nolan[1], Frédérique Vallières ●[1]\*, Mairéad Finn ●[1,3]**

**1** Trinity Centre for Global Health, Trinity College Dublin, Dublin, Ireland, **2** Food and Nutrition Officer, Ministry of Agriculture, Irrigation and Water Development, Lilongwe, Malawi, **3** Department of Public Health and Epidemiology, Royal College of Surgeons in Ireland, Dublin, Ireland

\* fvallier@tcd.ie

## Abstract

### Background and aim

Despite adequate food production and nutrition intervention coverage, stunting remains an enduring problem in Ntchisi, Malawi. Globally, gender and social norms are known to influence nutritional outcomes in children. This study explores how gender norms contribute to child stunting, in Ntchisi district, Central Malawi.

### Research methods

Informed by the UNICEF Framework for Malnutrition, nine focus group discussions were conducted with a target population of mothers (n = 24), fathers (n = 23) purposively targeted through growth monitoring sessions, and members of policy and health treatment committees (n = 21), spanning three different areas of Ntchisi district. Data were analysed through inductive thematic analysis, guided by the framework for Research in Gender and Ethics (RinGs).

### Results

Three primary themes were identified: 1) gender unequal decision making on the consumption, sale and distribution of food; 2) enshrined community norms influence feeding practices underpinned by gender-based violence; and 3) policy disconnections and gaps that reinforce gender norms regarding nutrition. Themes encompassed practices across household, health treatment, and policy level.

### Conclusion

Gender norms that underpin inequalities in decision making for production and consumption of food undermine children's nourishment and contributes towards sustained child malnutrition in Ntchisi. Existing policy documents should revise their guidelines to incorporate gender norms as key determinants of malnutrition.

**Data Availability Statement:** Data relevant to this study are available from Zenodo at https://doi.org/10.5281/zenodo.11478993.

**Funding:** The author(s) received no specific funding for this work.

**Competing interests:** The authors have declared that no competing interests exist.

## Introduction

The World Health Organization (WHO) defines malnutrition as "deficiencies, excesses or imbalances in a person's intake of energy and/or nutrients" [1] broadly incorporating under-nutrition, micro-nutrient related malnutrition, and over-nutrition. Malnutrition remains a topic of global concern, with 20% of the global under-five population (149.2 million) currently 'stunted,' that is, a low height-for-age index, [2] and where malnutrition accounts for 53% of under-5 mortality [3]. Stunting causes long-term impairment including poor school performance in childhood, reduced productivity later in life, and an increased risk of chronic non-communicable illnesses; and, in the worst cases, death [4].

Numerous international policy commitments seek to address malnutrition. The United Nations (UN) Sustainable Development Goals (SDGs), particularly SDG2, aspire to improve agriculture and food systems to feed and nourish malnourished populations [5]. Likewise, the WHO's 2018 Global Nutrition Target (GNT) sought to reduce the number of stunted children by 40% [6] and the UN's 2015 General Assembly declared 2015–2025 to be the decade of nutrition, a commitment by member states to implement policies, programs, and investments to eliminate malnutrition in all its forms. Scaling Up Nutrition (SUN)–a global movement working with the UN, civil society, business and donor networks to tackle malnutrition—promotes exclusive care and investment in child care globally, particularly in the first 1000 days of life [7].

Established determinants of malnutrition include poor maternal nutrition and young child feeding practices [8,9]; young mothers' age at first birth, which is linked to higher parity and poorer birth spacing practices [10]; depleted nutrition for mothers experiencing successive pregnancies and lactation [8], associated with poorer feeding practices [11]; subclinical infections from exposures to contaminated environments [12,13]; and infectious diseases such as malaria, which can compound short term loss of weight and contribute to long term stunting [14]. Malnutrition is thus determined by the complex interaction of multiple social, economic, and cultural factors. These factors are reflected in UNICEF's Conceptual Framework of Maternal and Child Nutrition (Fig 1) [15], often considered the primary global policy instrument to address the causes and consequences of malnutrition and reduce its prevalence [16]. According to this framework, *basic-level* causes operate in the broader context and include socio-economic and cultural context, human capital, and access to potential resources; *underlying* causes operate at household and community level, including care and feeding practices; and *immediate* causes relate to direct diet intake and the presence of disease[16]. Recently updated in 2020, the framework now also recognizes contributors to good nutrition as being either immediate, underlying, or enabling determinants [17].

Gender refers to the socially constructed roles, behaviors, activities, and attributes that a given society considers appropriate for men and women [18]. More recently, growing attention has been brought to the impact of gender norms on child malnutrition and stunting [19]. In many settings, women still cannot exercise freedom and autonomy of choice, including within their own households [14], with the extent to which a mother can make independent decisions negatively associated with the likelihood of childhood stunting [19]. Similarly, granting women greater decision-making power is associated with notable improvements in their children's height-for-age index [20]. Likewise, analysis of gender relations in norms and decision-making has recently come to the fore in explaining the endurance of problematic health outcomes [21]. Indeed, positive social and cultural norms (which include gender) are recognized as one of the most enabling determinants of improved nutrition for women and children [17].

In recognition of the role of gender and social norms in determining nutritional outcomes in children, several frameworks have been created to guide gender analyses within health and nutritional programming [22–24], which differ in their focus on types of power or area of the

Reinhardt and Fanzo

Addressing chronic malnutrition

|The UNICEF conceptual framework of undernutrition is shown. Source: UNICEF. *Improving Child Nutrition: The achievable imperative for global progress*. United Nations Children's Fund; 2013. p. 4.

**Source**: Reinhardt K, Fanzo J. Addressing Chronic Malnutrition through Multi-Sectoral, Sustainable Approaches: A Review of the Causes and Consequences. Front Nutr. 2014;1:13.

**Fig 1. UNICEF framework for malnutrition.**

health system. Specifically, the Research in Gender and Ethics (RinGS) Framework articulates gender differences as unfolding across four domains: access to resources; division of labour, roles, and everyday practices; social norms and values; and rules and decision making [25,26]. It is considered comprehensive for examining social and structural elements of gender interacting with health [21].

## Malnutrition in Malawi

Categorized as one of the poorest countries in the world by the World Bank [27], Malawi is a Southern African country home to over 18 million inhabitants. A signatory to the SUN movement since 2011, the Malawian Government applies UNICEF's Conceptual Framework for Malnutrition (1990) to address malnutrition [16]. Moreover, the Malawi Multi-Sectoral National Nutrition Policy (MSNNP) sets out the country's priority areas for implementing nutrition interventions across the country [28]. Specifically, these priority areas include (i) the prevention of under-nutrition, (ii) achieving gender equality, equity protection, participation and empowerment, and (iii) the treatment and control of malnutrition [28]. The MSNNP

outlines best practices to end malnutrition, detailing policy and implementation plans to address causes of malnutrition at household, health treatment, and policy levels [28].

Malawi has registered some reduction in the prevalence of stunting among children under five from 47% in 2010 [29], to 37% in 2015 (from the most recent available data), which is still very high. One of the most poorly affected districts in the country is Ntchisi, where 40% of the population are malnourished [29]. Paradoxically, Ntchisi is also considered one of the most household food secure districts in the country [30,31]. Being food secure defined as people having at all times physical, social and economic access to food, safely consumed in sufficient quantity and quality for dietary needs and food preferences, supported by an environment of adequate water and sanitation, health services and care [32,33]. Furthermore, exclusive breast-feeding rates in Ntchisi district for infants under six months (61%) exceed the global average (40%) [2,29].

Evidence suggests that gender norms may play a role in Ntchisi's observed malnutrition rates. For example, in 2016, 47% of women were married before 18 years and 29% of adolescent girls were mothers [29,34]. Women were also found to be five times less likely to make independent decisions when compared with men, a difference linked to variation in earning power [35]. There is evidence that differences in gender norms have a detrimental impact on women's health [36]. *How* differences in gender norms may contribute to malnourishment and stunting in children, in Ntchisi, Malawi, however, remain unexplored. This study therefore aimed to explore how gender norms play a role in continued stunting in Ntchisi District, Malawi, and to contribute to a revised version of the UNICEF Framework for Malnutrition to better reflect gender norms.

## Materials and methods

### Design and sampling

We employed a qualitative design, as part of a wider examination of the causes of observed stunting in Ntchisi district. We conducted nine focus group discussions (FGDs) to explore the role of gender norms in persistent child stunting across three different communities. FGDs were selected as they are considered valuable for generating data on group social norms, while also prioritising participants' perspectives [37]. Each community was purposely selected to achieve geographical spread. Six FGDs—two in each community—were conducted with mothers and fathers as household members, one FGD was conducted with participants delivering nutrition health treatment and two were conducted with participants working on nutrition policy. Therefore, nine focus groups explored how gender norms might impact (mal)nutrition, reflecting household, health treatment and nutrition policy spheres as articulated in the Malawi MSNNP [28].

In total, 68 participants were recruited, with 6–9 people represented within each FGD. The six FGDs with household members comprised three with males (n = 24) and three with females (n = 23). Females were within the reproductive age group, had at least one child under the age of five in the household, and were recruited through Growth Monitoring Sessions. There was no age limitation for males, who were also required to have at least one child under the age of five in the household. Men were recruited from nearby social gathering venues, mostly at church choir gatherings and market points via word-of-mouth by the government extension workers.

At health treatment level, participants (n = 10) included five females and five males of the Nutrition Rehabilitation Unit (NRU) Management Committee. This committee comprises nutritionist, nurse, and home-help workers providing treatment to severe cases of malnutrition admitted to the District Health Office. Policy level participants (n = 10) also comprised

five females and five males of the District Agriculture Coordinating Committee (DAECC), a committee that oversees and coordinates all agricultural interventions at the district level, and the District Nutrition Coordinating Committee (DNCC), a committee responsible for coordinating all nutrition interventions at the district.

## Data collection instruments

Semi-structured focus group discussion guides were developed with reference to the domains of the UNICEF Framework of Malnutrition [16,17], incorporating probes to ensure maximum investigation of persistent stunting. The guides included general questions about daily activities, food and income; resources and decision-making; and understanding of and perspectives on stunting. Discussions were conducted by the first author in Chichewa, the local language. Two field assistants, one from the health sector and the other from the agriculture sector, took notes of the participants' answers throughout the discussions as a backup for the audio recordings.

## Data management and analysis

FGDs were recorded, transcribed verbatim [38], and translated from Chichewa to English with close reference to audio files and notes. All personal identifiers were removed. Following a thorough review of the transcripts for familiarity, analysis comprised three phases of open (i.e. inductive) coding, categorization, and abstraction. Transcripts were coded according to the steps of Braun and Clarke [39], with common phrases assigned codes, then inductively organized into categories and themes. A framework for gender analysis [25,26], informed by Research in Gender and Ethics (RinGs) [22] guided analytic insights. Interpretation of the data was enhanced by discussions between the first author, the field workers, and the second author to refine analytical development and verify the conceptual coherence of the codes.

## Ethics, validity and reliability

Ethics approval was granted by the Trinity Health Policy and Management, Centre for Global Health (HP&M/CGH) Research Ethics Committee (REC) (Ref. 20/2029/01) and by the Ntchisi District Council Secretariat (Ref 2019/01/01). Informed verbal consent was sought from participants, who were fully briefed about the study aims and objectives, the voluntary nature of their participation, and that they could opt out at any time without penalty.

The validity and reliability of the data was ensured through transcription, clear and transparent coding, reflexivity, and cross-checking of themes between the lead author, the second author, and two field assistants [40]. The first author, who carried out the data collection, is a nutritionist in Ntchisi District, and reflexive attention was paid to the influence of her positionality throughout the data collection and analysis phases.

## Results

### Participant characteristics

Discussions commenced with a brief introduction from all participants. From this, it was gleaned that female participants ranged between 18 to 38 years of age and males ranged from 25 to 56 years of age. Participants reported having five children, on average. From the introductions, it was also gleaned that most mothers and fathers were educated to primary level only, with some to second level. Agriculture was their main source of food and income. Among committee members, half were employed by government while the other half were employed by non-government organizations.

| | Domains of RinGS Framework for Gender Analysis | Corresponding Theme | Categories |
|---|---|---|---|
| **How gender norms affect stunting in Ntchisi district, Malawi** | *Division of labour, roles and everyday practices*<br><br>*Rules and decision making* | Gender unequal decision making on the consumption, sale and distribution of food | Distinction of roles in IYCF |
| | | | Competing spending needs |
| | | | Nutrition not a priority in household spending |
| | | | Men control decisions on use of farm produce |
| | | | Mothers at health centres, fathers at home |
| | *Social norms and values* | Enshrined community norms influence feeding practices underpinned by gender based violence | Household conflict and gender based violence |
| | | | Women accept GBV: fear and avoid shame |
| | | | Prevalence of GBV after harvesting time |
| | | | Men's status in the community an enduring norm |
| | *Access to resources* | Policy disconnections and gaps that reinforce unequal gender norms at the structural level | Agriculture, nutrition interventions disconnected |
| | | | Poor male involvement in nutrition interventions |
| | | | Teen motherhood and large family sizes |
| | | | Low education and literacy levels |

**Fig 2. Gender norms contributing to stunting in Ntchisi District, Malawi.**

## Exploring the role of gender norms in contributing to malnutrition

Three core themes were identified in the data to illuminate the role of gender norms in contributing to malnutrition and stunting in Ntchisi. These were: (i) gender unequal decision making on the consumption, sale and distribution of food; (ii) enshrined community norms influence feeding practices, underpinned by gender based violence; (iii) policy disconnections and gaps that reinforce unequal gender norms regarding nutrition. Themes are presented in Fig 2, aligned to the RinGS Framework for Gender Analysis [21,22].

## Gender unequal decision making shapes the household production and consumption of food

Power differences between fathers and mothers affected decisions about how farm produce is utilized. The policy level focus groups explained how practices in subsistence agriculture affect what to produce, the quantity to produce, and where to sell produce, all of which affect how much remains for families to eat. Women in the district were reported to have lower decision-making power about food production by comparison to men:

> "Men dominate. . .they control what is cultivated and what to do with the produce. You find that a woman might know that legumes are nutritious and essential for child growth. . . [but] when the husband wants to sell, the [wife] will not resist." DAECC FGD.

Describing their social context, household participants told of the different roles that mothers and fathers take and how this impacts children's nutrition. The participants recounted that within the household, fathers are considered providers while mothers are nurturers. When describing their daily routine, mothers mentioned household chores like cooking, feeding, laundry, tidying the house, and collecting firewood and water, the latter usually from a great

distance. On the other hand, fathers reported daily work of gardening, casual tasks, socializing, and sourcing income for the household. Fathers and mothers emphasised a clear distinction of roles in feeding children:

"Men search for food. . .we stay at home taking care of the children. . . cooking for and feeding them. When men bring food, we cook." Mothers FGD

"We take care of children in the sense that we search for food so that woman can cook it for the children." Fathers FGD.

However, fathers were characterized by participants as not health or nutrition minded in relation to expenditure. Almost all mothers' groups described that men in the community use their income on pleasures rather than on actual needs.

"Sometimes, you can work hard to produce more. However, when you harvest and sell, the man will spend it on drinking and other women. You cannot question how they spend it. They forget about their own children. You face it all because the children usually stay at home with a mother". Mothers FGD.

Similarly, fathers indicated that when they harvest, they have several responsibilities to address. Nutrition might not usually be the priority.

"A poor person always has a handful of things to take care of. When you get money. . .you don't remember what nutrition or nutritious food is about" Fathers FGD.

Differences between mothers and fathers in decision making about utilization of farm produce also manifested at the health center. The health committee FGD recounted how in-patient cases of malnutrition at the health centre are presented mostly by mothers, who attend with their children while fathers stay at home. Extended stays can cause worry to mothers about further disruption to home life and the oversale of food:

"The mothers wait for children here. . .we give them health talks every morning. Fathers usually stay at home. . .[but]. . .when the mothers stay for some days, they start complaining thinking that their husbands will marry another woman or get a chance to sell out farm produce and other household assets. They just become so eager to go, and it is evident in how they act." NRU FGD.

Although mothers in Ntchisi were primarily responsible for feeding children, they lacked decision-making power over the production and consumption of food. Fathers, who held decision making power, were not oriented towards nutritional needs for children. Furthermore, inequality in decision making about food for children intensifies when women are not present.

## Enshrined community norms influence feeding practices, underpinned by gender based violence

Household focus groups reported that inequity in household decision making around food was preserved by traditional norms of gender-based violence (GBV) in Ntchisi district. A clear and strong description by participants was that women cooperate with men's decisions for fear of GBV, and do not speak out about GBV in fear of shaming their husbands and losing respect.

"In this area, when a woman has a brilliant idea, men do not want to appear cheap, so they do not agree, whatever it is about. . .as a woman you just coil up and keep quiet otherwise you will be beaten" Mothers FGDs

The supremacy of this social norm was bolstered by the status it confers on men in the community.

"Men here say things, and that is final. . .maybe it could be one out of ten men that would be under petticoat government. If you allow your wife to be key in making decisions, your friends will despise you. . .even young children would look down on you." Fathers FGD.

District committee members recounted that cases of GBV are more prevalent at times of the year when produce is sold. Gaps in power between mothers and fathers extended to a firm inequality in autonomy, which impacted on children's nutrition.

"Men are just bad. During the cultivating period, they work well with their wives, but when it is selling time, they prefer to work alone and spend the money on drinking or other women. . .this initiates conflicts that usually ends in abuse. Children are usually left with mothers that do not have food supplies or money." DNCC FGD.

Mothers and fathers highlighted that the clusters of friends and neighbours they lived among were their primary sources of knowledge, indicating the strong traditional norms shaped how children were taken care of. Gender based violent practices were encased in traditional community norms.

"Everybody does that. . .that's how we do things here. Our parents teach us what they did so that we grew up. When your child is born, they take you aside and teach you many things. . . .we apply that so that our children grow up too" Fathers FGD.

Thus, community adherence to traditional practice creates a challenge to do things differently when feeding children. Rigid social norms of women required to be deferential to men were held in place by traditional expectations and the threat of gender-based violence. These norms retained an enduring social power in the community and impacted practices and decision making around food for children.

### Policy disconnections and gaps that structurally reinforce gender norms regarding nutrition

A disconnection between production and nutrition interventions was articulated by policy level participants, with agricultural produce sold by subsistence farmers rather than retained for their families. Participants explained that while the agriculture committee incorporates nutrition interventions to its activities in theory; no nutrition achievements could be attributed to this work.

"We try to incorporate nutrition in our planning, but all detailed plans and achievements are with the DNCC because we mainly focus on agriculture . . . however, nutrition is for DNCC, so consult them for specific achievements." DAECC FGD.

The disconnection indicates a wider aspect of the gender gap because agriculture interventions—usually organized by men in the district—are implemented separately from general nutrition interventions, targeting women as primary caregivers for children.

Men's participation in the implementation of nutrition interventions was noted as a gap by household level participants. Mothers encouraged targeting fathers for inclusion, highlighting that if men were aware of nutrition issues, they would be more supportive at home and more informed about nutrition-related decisions.

"Men should be taught. They are stubborn and irresponsible because they do not know and understand. They should hear for themselves. They should be included in care groups. Now care groups are only full of women" Mothers FGD

Despite father's lack of involvement in nutrition reflecting the socio-cultural norms in Ntchisi, men in the fathers' focus groups indicated their willingness to participate in the sessions and learn more about nutrition.

"The extension workers always teach women. The next training, please include us. . .we will come and are happy to learn." Fathers FGD

Impacts of trainings were also affected by education levels, specifically literacy, of the population. Low levels of education were reported as a general feature of the population in Ntchisi, affecting practices towards childcare. Health treatment level participants highlighted how literacy levels affect the uptake of trainings.

"You can have a hospital, offer training in all these things. . .however, without school, it is all in vain. You need some [education] to understand the importance of it all." NRU FGD.

The Government's role was further emphasized in reducing teen pregnancies and achievements in nutrition education. Teen mothers face challenges in adequately nourishing themselves and their children. Pressure is exacerbated when women have their first child at a young age, because it leads to larger family size. Care demands of young children cannot be met if the mother has many children, closely spaced together.

"I think the issue is that people have many children. You see some have eight, others nine. They cannot afford to monitor if all ate or not. You would not know that one missed food because they are too many..[. . .]. . . the Government should do more in reducing teen pregnancies. . .and should invest in keeping girls in school." NRU FGD.

Low levels of education undermine the household's ability to meet the nutrition needs of children, affecting both choices in health and well-being and the priority given to well-being. These trends are also an indication of a broader structural context with unequal gender norms impacting women and children.

## Discussion

Despite a context of food security, interventions on nutritious feeding, and the development of policies to target the eradication of undernutrition, children's inadequate consumption of food in Ntchisi is impacted by gendered practices and power dynamics. Specifically, differing roles between men and women shape use of farm produce and feeding practices, enduring traditional social norms preserved via the threat of gender-based violence and structurally bolstered through policy disconnections, are all experienced as gender-related factors that impact on the nutrition of children in Ntchisi.

Similar to Alemayehu et al.[14], and Kamiya et al.[41], women interviewed in Ntchisi district also had limited power in making decisions about use of food within their households. Adhering to men's decisions regarding food production is a strong tradition that results in households being unable to consume the right varieties and quantities. Although subsistence farming is generally characterized by the consumption of produce [42], the narratives in this study highlighted how food produced for subsistence is sold and distributed rather than consumed. Subsistence farming also connotes adherence to indigenous knowledge and traditions in production and consumption, pointing again to the embedded nature of traditional practices [43].

Power differences between women and men were also evident in that women were involved in harvesting food, but excluded from decisions on the use of the harvest. Conflict related to decisions on use of harvest produce reportedly resulted in increased gender-based violence or a rupture in the marriage, as participants recounted risks of both increased violence and fathers sometimes abandoning their families. This trend has been identified elsewhere, where women characterized themselves as 'harvest wives' in cases where their husbands abandoned them when the harvest was complete [44]. A salient phenomenon that impacts the use of food is women's lack of inclusion in decisions on how harvested food will be used for the family, with a heightened risk of gender based violence or relationship breakdown at this point.

While the government has an established role in imparting knowledge on food preparation and feeding, enshrined gender norms undermine how women can practice their learning, as they are pressured to adhere to men's decisions. Women's translation of their nutrition knowledge into practice is also challenged by the high number of GBV cases reported in the district, reported as more prevalent during harvest time, also contributing to the limited contribution of agricultural produce to nourishment of children. Consistent with previous findings, children's dietary intake in Ntchisi is also reportedly further compromised by mothers' young age and a large number of children, with consequent reduced care practices [45]. In the context where women do not make independent decisions, it can be assessed that the lack of independence extends to decisions on family size and birth control methods.

Father's lack of involvement in their child's nutrition within Ntchisi was also found to be a manifestation of the social cultural norms in the district. Participants across the sample suggested inclusion of men in community trainings on infant and young child feeding. In a social context where fathers want to be respected for not being "under petticoat government", the cultural change required to incorporate men into nutrition interventions is evident. A more targeted approach to addressing men's roles in nutrition, that takes account of the bearing of gender norms, is required.

In support of the UNICEF Conceptual Framework of Malnutrition, the findings from this study demonstrate that gender dynamics indeed shape malnutrition across all three causes of malnutrition articulated in the framework—basic, underlying, and intermediate. Practices at the underlying levels of household and community related to the production and consumption of food were highly unequal for men and women. At the basic level of socio-economic and cultural context, capital, and access to resources, powerful norms bolstering inequity between women and men were evident. Unequal practices within these domains were highly gendered, intersecting to shape lack of access to and use of resources for the nourishment of children. These findings and considerations are presented in Fig 3 as an expanded version of the UNICEF Conceptual Framework for Maternal and Child Malnutrition.

The current study adds to the existing literature by illustrating that in Ntchisi, gender norms manifest across basic, underlying, and intermediate levels, and not just at the basic level as currently articulated within the UNICEF conceptualization. Accordingly, gender should be articulated across the entire UNICEF framework, and the framework might benefit from

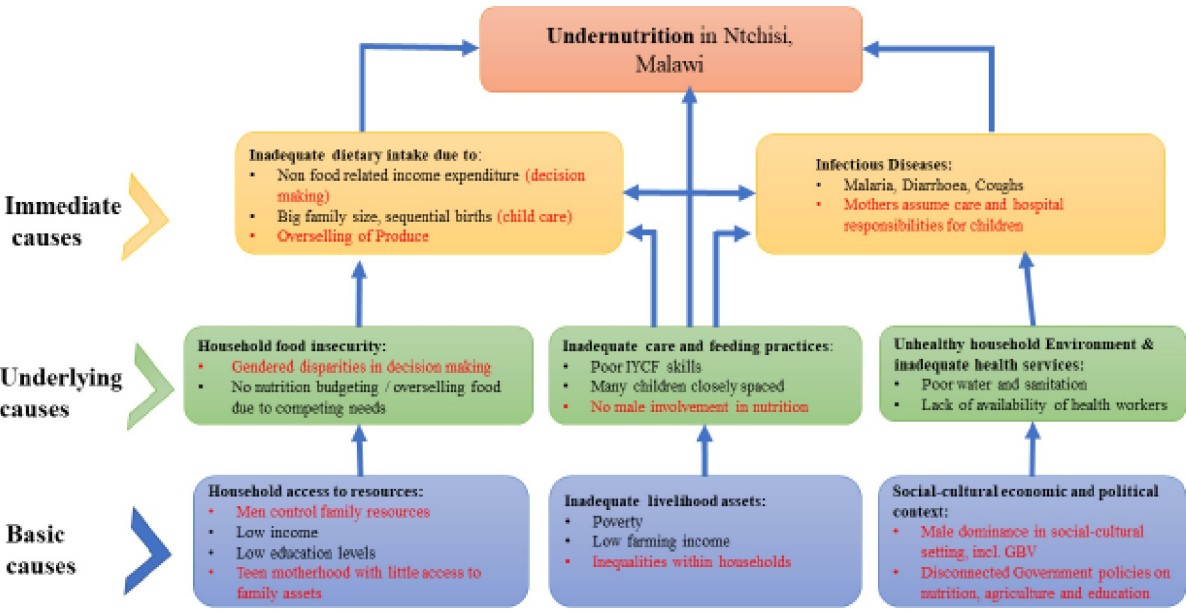

**Fig 3. Factors contributing to stunting adapted to the 1990 UNICEF conceptual framework of malnutrition.**

applying the WHO's recommendations for incorporating intersectional gender analysis [25,26]. In line with the domains within RiNGs framework, domains of 'access to resources'; 'divisions of labour, roles and everyday practices'; 'social norms and values'; 'rules and decision making' [26] should also be incorporated into the UNICEF framework to illuminate how unequal gender norms and roles unfold to explain continued problems of nutrition for children.

The lack of policy coordination between agriculture and nutrition could contribute to explanations of the disconnection between food production, food consumption, and nutrition status in Ntchisi. Strengthening policy linkages between these domains would align with the 2017 Global Nutrition Report [46] statement that achieving good nutrition means practicing agriculture that feeds and nourishes a nation state. The Food and Agriculture Organization emphasizes the importance of integrating food production and nutrition interventions in parallel, and the findings presented here provide insights for practical implementation [47,48]. Integrating nutrition and agriculture is emphasized by the theme of the 2016 World Food Day (WFD) commemorations; "our health is not only what we eat, it is primarily what we grow" [49]. Gender sensitivity must traverse these connections. Male involvement in these connections is imperative, in recognition of the fact that they hold decision-making power and asset ownership. Mkandawire & Hendricks (2015) note that while it is important to target women and children in nutrition interventions, the exclusion of men ignores the fact that women are limited in terms of decision-making powers and asset ownership [50].

The Malawi MSNNP cites gender equality, equity protection, participation and empowerment [28] as its second priority area. The findings can be translated into policy actions on under-nutrition and stunting under the umbrella of the MSNNP. Specifically, government and development partners could continue multi-sectoral implementation but place special focus on male involvement.

The Malawi MSNNP 2018–2022 emphasizes the need to involve all line ministries and stakeholders in reducing child under-nutrition. In line with the policy objectives, this study

recommends that Government and other NGOs connect programmes across household, health treatment and policy levels in Ntchisi; connect agriculture and nutrition polices; and incorporate a gender lens. In addition, education must underpin all interventions, especially so for girls experiencing teen pregnancy. The successful implementation of nutrition programmes relies on an educated and literate population.

## Conclusion

In conclusion, the findings of this study have demonstrated how (mal)nutrition is impacted by the socially constructed roles, behaviours, activities and attributes that a given society considers appropriate for men and women, which can manifest as unequal power relations shaping access to and use of food [26]. The unequal division of power and resources between men and women in Ntchisi district in the central region of Malawi is a contributing factor to levels of malnutrition and stunting. To address these influences on (mal)nutrition, global and national policy documents can integrate gender sensitive approaches cutting across other policy areas, allowing gender to be fully integrated in response. it appears appropriate to recommend that intersectional gender analysis that takes account of gender inequality at household and community levels, should be mainstreamed into the UNICEF framework as also recommended by the WHO. Efforts to improve child nutrition need to be aligned with gender interventions to improve access, utilization and consumption of nutritious foods. UNICEF's conceptual framework of malnutrition should highlight gender for more holistic programming to reflect that improving the nutrition status of children in a highly productive agricultural region is only possible if gender and nutrition interventions are indissolubly linked.

### Limitations

The research had several limitations. Firstly, interviews were conducted in Chichewa, the vernacular language in Ntchisi, and transcripts were translated to English by the first author who speaks Chichewa and English fluently. This translation process, however, might have introduced some mis-explanations of meanings of terms. Second, the first author is a nutrition officer who works in the area under study. While every effort was made to account for the researcher role, it is challenging to fully eliminate a researchers' influence on participant responses. In this case, the researcher's gender or role as a nutritionist may have influenced the discussions. Thirdly, data collection comprised FGDs with policy, health management and household members in the community, excluding interviews within individual households. The household level interviews would have created room to capture feeding pattern, which this study did not. Future research may want to explore this and how intra-household dynamics impact on the nutritional status of women. Future studies should also aim to examine the correlation between gender norms and children's nutritional status.

## Acknowledgments

We would like to thank the participants and fieldworkers who gave their time to this study, and without whom this research would not have been possible.

## Author Contributions

**Conceptualization:** Whitney Mphangwe, Mairéad Finn.

**Data curation:** Whitney Mphangwe.

**Formal analysis:** Whitney Mphangwe.

**Investigation:** Whitney Mphangwe.

**Methodology:** Whitney Mphangwe.

**Supervision:** Mairéad Finn.

**Writing – review & editing:** Ann Nolan, Frédérique Vallières, Mairéad Finn.

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
