## [Decision Letter · Decision Letter 0]

7 May 2024

PONE-D-23-21165How do gender norms contribute to stunting in Ntchisi District, Malawi? A qualitative studyPLOS ONE

Dear Dr. Vallières,

Thank you for submitting your manuscript to PLOS ONE. After careful consideration, we feel that it has merit but does not fully meet PLOS ONE’s publication criteria as it currently stands. Therefore, we invite you to submit a revised version of the manuscript that addresses the points raised during the review process.

**ACADEMIC EDITOR: I am pleased to inform you that experts in the field have reviewed your manuscript. You're to address their comments or suggestions as early as possible.==============================**

**Please submit your revised manuscript by 1**Jun 21 2024 11:59PM; 11:59PM. If you will need more time than this to complete your revisions, please reply to this message or contact the journal office at plosone@plos.org. **Please include the following items when submitting your revised manuscript:****A rebuttal letter that responds to each point raised by the academic editor and reviewer(s). You should upload this letter as a separate file labeled 'Response to Reviewers'.****A marked-up copy of your manuscript that highlights changes made to the original version. You should upload this as a separate file labeled 'Revised Manuscript with Track Changes'.****An unmarked version of your revised paper without tracked changes. You should upload this as a separate file labeled 'Manuscript'.******If applicable, we recommend that you deposit your laboratory protocols in protocols.io to enhance the reproducibility of your results. Protocols.io assigns your protocol its own identifier (DOI) so that it can be cited independently in the future. For instructions see: https://journals.plos.org/plosone/s/submission-guidelines#loc-laboratory-protocols. Additionally, PLOS ONE offers an option for publishing peer-reviewed Lab Protocol articles, which describe protocols hosted on protocols.io. Read more information on sharing protocols at https://plos.org/protocols?utm_medium=editorial-email&utm_source=authorletters&utm_campaign=protocols**.

**We look forward to receiving your revised manuscript.**

**Kind regards,**

**Olutosin Ademola Otekunrin**

**Academic Editor**

**PLOS ONE**

3. In the online submission form, you indicated that [The pseudo-anonymised qualitative data is available from the first author, upon reasonable request.]. 

**4. Please review your reference list to ensure that it is complete and correct. If you have cited papers that have been retracted, please include the rationale for doing so in the manuscript text, or remove these references and replace them with relevant current references. Any changes to the reference list should be mentioned in the rebuttal letter that accompanies your revised manuscript. If you need to cite a retracted article, indicate the article’s retracted status in the References list and also include a citation and full reference for the retraction notice**.

**Additional Editor Comments**:

****

**Reviewers' comments**:

**Reviewer's Responses to Questions**

**Comments to the Author**

1. Is the manuscript technically sound, and do the data support the conclusions?

**The manuscript must describe a technically sound piece of scientific research with data that supports the conclusions. Experiments must have been conducted rigorously, with appropriate controls, replication, and sample sizes. The conclusions must be drawn appropriately based on the data presented. **

**Reviewer #1: Yes**

**Reviewer #2: Yes**

**2. Has the statistical analysis been performed appropriately and rigorously? **

**Reviewer #1: Yes**

**Reviewer #2: Yes**

**3. Have the authors made all data underlying the findings in their manuscript fully available?**

**The PLOS Data policy requires authors to make all data underlying the findings described in their manuscript fully available without restriction, with rare exception (please refer to the Data Availability Statement in the manuscript PDF file). The data should be provided as part of the manuscript or its supporting information, or deposited to a public repository. For example, in addition to summary statistics, the data points behind means, medians and variance measures should be available. If there are restrictions on publicly sharing data—e.g. participant privacy or use of data from a third party—those must be specified.**

**Reviewer #1: Yes**

**Reviewer #2: Yes**

**4. Is the manuscript presented in an intelligible fashion and written in standard English?**

**PLOS ONE does not copyedit accepted manuscripts, so the language in submitted articles must be clear, correct, and unambiguous. Any typographical or grammatical errors should be corrected at revision, so please note any specific errors here.**

**Reviewer #1: Yes**

**Reviewer #2: Yes**

**5. Review Comments to the Author**

**Please use the space provided to explain your answers to the questions above. You may also include additional comments for the author, including concerns about dual publication, research ethics, or publication ethics. (Please upload your review as an attachment if it exceeds 20,000 characters)**

**Reviewer #1: Congratulations to the authors. The paper is an interesting read and is well written in a scientific manner. The paper addresses an important subject matter of malnutrition and gender roles are indeed a mater issues that needs to be addressed**

**Reviewer #2: REVIEWERS COMMENTS**

**Generally, the study titled “How do gender norms contribute to stunting in Ntchisi District, Malawi? A qualitative study” was well-researched, very relevant, scientific, and very informative, it raised pertinent issues regarding the impact of gender norms on stunting in children.**

**The authors demonstrated excellent mastery of the study. However, few minor corrections are needed before the manuscript proceeds for publication.**

**Please find below the area that needs clarification and improvement.**

**In the abstract, the study appears not to have target population.**

**Lines 48& 49 Defined by the World Health Organization (WHO) as “deficiencies, excesses or imbalances in a person’s intake of energy and/or nutrients”- What was defined by WHO? The statement needs to be reconstructed.**

**Line 112 While reduced from 47% in 2010,(29) the prevalence of stunting in children under-five years in………………..**

**Line 113 . Among one of the most poorly…………………**

**Lines 181-183 Ethics approval was granted by the Trinity Health Policy and Management, Centre for Global Health (HP&M/CGH) Research Ethics Committee (REC) and by the Ntchisi District Council Secretariat, including the District Health Office (DHO) and the District Agriculture Office (DAO). Informed verbal……………**

**Line 199 a mean age of 26 while fathers had a mean age of 38. Participants reported………………**

**Lines 200 & 201 Most mothers and fathers were educated to primary level only, with some to second level.**

**Line 332 Discussion and Conclusion**

**Lines 439 till end References- effect needed corrections as indicated.**

**Summary: The topic and the discuss appears not to tally, there are no information on feeding pattern of the children and their nutritional status. Secondly, is the study a purposive sampling of parents with stunted children? Thirdly, there is no correlational study to show the impact of gender norm on children’s nutritional status, so the study appears to be inconclusive.**

**6. PLOS authors have the option to publish the peer review history of their article (what does this mean?). If published, this will include your full peer review and any attached files.**

**Reviewer #1: No**

**Reviewer #2: No**

****

**While revising your submission, please upload your figure files to the Preflight Analysis and Conversion Engine (PACE) digital diagnostic tool, https://pacev2.apexcovantage.com/. PACE helps ensure that figures meet PLOS requirements. To use PACE, you must first register as a user. Registration is free. Then, login and navigate to the UPLOAD tab, where you will find detailed instructions on how to use the tool. If you encounter any issues or have any questions when using PACE, please email PLOS at figures@plos.org. Please note that Supporting Information files do not need this step.**

---

## [Author Response · Author response to Decision Letter 0]

4 Jun 2024

REVIEWERS COMMENTS

Reviewer 1: Congratulations to the authors. The paper is an interesting read and is well written in a scientific manner. The paper addresses an important subject matter of malnutrition and gender roles are indeed a mater issues that needs to be addressed 

Respnse: Thank you for your time and consideration in reviewing this piece. We appreciate your feedback and believe that the manuscript is greatly improved as a result of the reviewer comments. Thank you again for your time in reviewing this.

Reviewer 2

Reviewer 2; In the abstract, the study appears not to have target population. 

Response: Thank you for this feedback. At community level, the target population was fathers and mothers at household level. At district/implementation level, the target population were the coordinating committees for nutrition and agriculture. At health management level, the target population was the NRU for health management. These have now been summarized and reported in the abstract under research methods.

Reviewer 2: Lines 48& 49 Defined by the World Health Organization (WHO) as “deficiencies, excesses or imbalances in a person’s intake of energy and/or nutrients ”- What was defined by WHO? The statement needs to be reconstructed. 

Response: Thank you for this observation, we have amended this.

Reviewer 2: Line 112 While reduced from 47% in 2010,(29) the prevalence of stunting in children under-five years in……………….. This phrase might not be appropriate to start a paragraph, please kindly re-phrase 

Response: Thank you for your observation and feedback. This statement has been rephrased. It is now on line 116.

Reviewer 2: Line 113. Among one of the most poorly………………… You might need to delete among

Response: Thank you for this observation, we have amended this. This phrase is now on line 119.

Reviewer 2: Lines 181-183 Ethics approval was granted by the Trinity Health Policy and Management, Centre for Global Health (HP&M/CGH) Research Ethics Committee (REC) and by the Ntchisi District Council Secretariat, including the District Health Office (DHO) and the District Agriculture Office (DAO) Insert the study’s ethical approval number 

Response: Reference Numbers have been added for both RECs.

Reviewer 2: Informed verbal…………… Thank you for this observation, we have amended this.

Reviewer 2: Line 199 a mean age of 26 while fathers had a mean age of 38. Participants reported………… Mean must be reported with its SD value 

Response: This information was gleaned from the focus group discussions rather than calculated based on the use of a demographic questionnaire. We have amended the language to reflect this in the manuscript.

Reviewer 2: Lines 200 & 201 Most mothers and fathers were educated to primary level only, with some to second level. What’s the percentage figure?

Percentage should also be reported to quantify the difference

Response: Thank you. This information was also gleaned from the brief introductions by the participants at the outset of the focus group discussions. The manuscript has been amended to reflect this on line 202.

Reviewer 2: Line 332 Discussion and Conclusion Conclusion must be separate as contained in the author’s guide for this journal

Response: Thank you for noting this. This has been amended. 

Reviewer 2: Kindly insert the DOIs for the cited articles, and effect all necessary corrections by providing the missing information in the references 

Response: Thank you for this feedback. This has been amended as reflected in the references section. The Vancouver style as stipulated by PLOS ONE does not include DOIs. We have adjusted to another numeric style, JAMA, which does include the DOIs. 

Reviewer 2: The topic and the discuss appears not to tally, there are no information on feeding pattern of the children and their nutritional status. 

Response: Thank you for this feedback. The discussion focused on the three main themes in the findings; decision-making around production and consumption, norms around feeding practices and the policy gaps fostering gender norms as fully presented in the results section. The research did not focus discussion around feeding patterns as children were accessed through growth monitoring sessions. These sessions screen children for malnutrition; on which data was already presented demonstrating that the levels of malnutrition are generally high in this area. The absence of information on feeding patterns has been added as a limitation in this study and a recommendation for future studies that research is conducted at household level to capture feeding patterns.

Reviewer 2: Secondly, is the study a purposive sampling of parents with stunted children? 

Response: Thank you for this observation. This study purposively targeted mothers and fathers at growth monitoring sessions in an area where children were generally stunted, as well as coordinating/management level committees. The first author was a district nutritionist and was familiar with the general experience of under-nutrition as captured through the growth monitoring sessions.

Reviewer 2: Thirdly, there is no correlational study to show the impact of gender norm on children’s nutritional status, so the study appears to be inconclusive.

Response: We appreciate this feedback and the time you took to review this document. This study was a qualitative investigation of the way in which gender roles shape access to resources and impact child stunting. Rather than defining and measuring impact, qualitative studies seek to highlight the processes by which particular phenomena occur and the meaning ascribed to that phenomena by participants. We have added the lack of correlation as a limitation and recommended that future studies should seek to examine this.

---

## [Decision Letter · Decision Letter 1]

25 Jun 2024

How do gender norms contribute to stunting in Ntchisi District, Malawi? A qualitative study

PONE-D-23-21165R1

Dear Dr. Vallières,

We’re pleased to inform you that your manuscript has been judged scientifically suitable for publication and will be formally accepted for publication once it meets all outstanding technical requirements.

Kind regards,

Olutosin Ademola Otekunrin

Academic Editor

PLOS ONE

Additional Editor Comments (optional):

Reviewers' comments:

Reviewer's Responses to Questions

**Comments to the Author**

1. If the authors have adequately addressed your comments raised in a previous round of review and you feel that this manuscript is now acceptable for publication, you may indicate that here to bypass the “Comments to the Author” section, enter your conflict of interest statement in the “Confidential to Editor” section, and submit your "Accept" recommendation.

Reviewer #1: All comments have been addressed

Reviewer #2: All comments have been addressed

2. Is the manuscript technically sound, and do the data support the conclusions?

Reviewer #1: Yes

Reviewer #2: Yes

3. Has the statistical analysis been performed appropriately and rigorously? 

Reviewer #1: Yes

Reviewer #2: Yes

4. Have the authors made all data underlying the findings in their manuscript fully available?

Reviewer #1: Yes

Reviewer #2: Yes

5. Is the manuscript presented in an intelligible fashion and written in standard English?

Reviewer #1: Yes

Reviewer #2: Yes

6. Review Comments to the Author

Reviewer #1: The authors have managed to address all the comments in a satisfactory manner. and I reccomend acceptance of the manucript for publication in your journal.

Reviewer #2: i will like to appreciate the authors for painstakingly effecting all the corrections and suggestions made. The manuscripts should proceed to production stage.

7. PLOS authors have the option to publish the peer review history of their article (what does this mean?). If published, this will include your full peer review and any attached files.

Reviewer #1: **Yes: **Dr Lesley Macheka

Reviewer #2: No

---

## [Editor Report · Acceptance letter]

19 Aug 2024

PONE-D-23-21165R1 

PLOS ONE

Dear Dr. Vallières, 

I'm pleased to inform you that your manuscript has been deemed suitable for publication in PLOS ONE. Congratulations! Your manuscript is now being handed over to our production team.

Kind regards, 

on behalf of

Dr. Olutosin Ademola Otekunrin 

Academic Editor

PLOS ONE